Bioinformatics profiling identifies seven immune-related risk signatures for hepatocellular carcinoma

Xue Feng 1
Yang Lixue 1
Dai Binghua 2
Xue Hui 1
Zhang Lei 1
Ge Ruiliang gerl_ehbh@sina.com 1
Sun Yanfu sunyanfu@hotmail.com 1
1 Department of Hepatic Surgery II, Eastern Hepatobiliary Surgery Hospital , Shanghai , China
2 Department of liver Transplantation, Eastern Hepatobiliary Surgery Hospital , Shanghai , China
Yang Xiang-Jiao
Electronic publication date: 2020 May 26
Publication date: 2020
Volume: 8
Electronic Location ID: e8301
Received 2019 Jul 12; Accepted 2019 Nov 26
Copyright: ©2020 Xue et al.
Copyright year: 2020
Copyright holder: Xue et al.
License: This is an open access article distributed under the terms of the Creative Commons Attribution License, which permits unrestricted use, distribution, reproduction and adaptation in any medium and for any purpose provided that it is properly attributed. For attribution, the original author(s), title, publication source (PeerJ) and either DOI or URL of the article must be cited.
License URL: https://creativecommons.org/licenses/by/4.0/

Keywords: Bioinformatics, Immunity, Prognostic markers, TCGA

Funding: Shanghai Medical Guiding Foundation 17411960300 The study was supported by the Shanghai Medical Guiding Foundation funding (17411960300) to Binghua Dai. The funders had no role in study design, data collection and analysis, decision to publish, or preparation of the manuscript.

==============================
Background

Density of tumor infiltrating lymphocytes (TIL) and expressions of certain immune-related genes have prognostic and predictive values in hepatocellular carcinoma (HCC); however, factors determining the immunophenotype of HCC patients are still unclear. In the current study, the transcript sequencing data of liver cancer were systematically analyzed to determine an immune gene marker for the prediction of clinical outcome of HCC.

Methods

RNASeq data and clinical follow-up information were downloaded from The Cancer Genome Atlas (TCGA), and the samples were assigned into high-stage and low-stage groups. Immune pathway-related genes were screened from the Molecular Signatures Database v4.0 (MsigDB) database. LASSO regression analysis was performed to identify robust immune-related biomarkers in predicting HCC clinical outcomes. Moreover, an immune gene-related prognostic model was established and validated by test sets and Gene Expression Omnibus (GEO) external validation sets.

Results

We obtained 319 immune genes from MsigDB, and the genes have different expression profiles in high-stage and low-stage of HCC. Univariate survival analysis found that 17 genes had a significant effect on HCC prognosis, among them, 13 (76.5%) genes were prognostically protective factors. Further lasso regression analysis identified seven potential prognostic markers (IL27, CD1D, NCOA6, CTSE, FCGRT, CFHR1, and APOA2) of robustness, most of which are related to tumor development. Cox regression analysis was further performed to establish a seven immune gene signature, which could stratify the risk of samples in training set, test set and external verification set (p < 0.01), and the AUC in both training set and test set was greater than 0.85, which also greater compared with previous studies.

Conclusion

This study constructed a 7-immunogenic marker as novel prognostic markers for predicting survival of HCC patients.

Introduction

As the most common malignant tumor in the liver, hepatocellular carcinoma (HCC) is the seventh most common cancer (Jemal et al., 2011) and the third leading cause of cancer-related deaths all over the world (Jemal et al., 2008). The recurrence rate of HCC within 5 years after surgery exceeds 70%, causing HCC-related mortality of about 80% (Bruix, Sherman & Practice Guidelines Committee AAftSoLD, 2005; Forner, Llovet & Bruix, 2012). The poor prognosis of patients with HCC is mainly resulted from the high progressiveness and metastasis of HCC and formation of new tumors in the diseased liver, which is the so-called “field effect” (Libbrecht et al., 2001; Sherman, 2008). More than 70% of HCC patients are unable to benefit from current treatments, due to potential liver dysfunction and advanced disease performance. The median survival time for patients with unresectable HCC is approximately 6 to 20 months, and their 5-year survival rate is less than 5% (Mikhail, Cosgrove & Zeidan, 2014). Therefore, it is highly urgent to discover prognostic biomarkers to accurately predict clinical outcomes of HCC patients.

Molecular profiles of tumor cells and cancer-related cells in their microenvironments are promising candidates for acting as predictive and prognostic biomarkers (Coussens & Werb, 2002; Gentles et al., 2015; Koscielny, 2010; Tarhini et al., 2017; Topalian et al., 2012). Though significant advances have been made in high-throughput sequencing technology, clinical application of these technologies has been hampered by different tumor types and heterogeneity (Dalton & Friend, 2006; Dupuy & Simon, 2007; Koscielny, 2010; Qi et al., 2016; Subramanian & Simon, 2010). Immune escape is a strategy, through which tumor cells evade host immune responses to continue their growth, and it is one of the hallmarks of human cancer (Corrales et al., 2017). Tumor immune escape is also considered as a promoter in the process of tumorigenesis and development, as it help tumor cells avoid or even block antigen stimulating immune response of tumor cells and provide appropriate microenvironment for tumor growth (Jiang et al., 2017; Kim, Emi & Tanabe, 2007). At present, studies have clarified the interaction between tumor and immune system (Angelova et al., 2015; Li et al., 2016; Rooney et al., 2015) and achieved remarkable progresses in advanced cancer treatment (Rooney et al., 2015); however, the treatment strategies only applies to a small number of patients.

Immunotherapy is another promising approach to overcome the poor prognosis after standard therapy as well as restricted applicability of targeted therapies to HCC patients. Currently, several immune-related parameters, mainly tumor infiltrating lymphocytes, have been reported to be able to predict the prognosis of HCC patients (Kim et al., 2018; Zhou et al., 2017), suggesting that immune states have a significant impact on the prognosis of HCC patients. Therefore, it is necessary to systematically study immunophenotype in HCC microenvironment, so as to better understand the complex anti-tumor response and help determine effective immunotherapy for HCC.

In the current study, expression profiles of HCC patients were obtained from large data sets from The Cancer Genome Atlas (TCGA) and Gene Expression Omnibus (GEO) databases, immune pathway genes were extracted from MsigDB, and different immune characteristics of HCC were analyzed through expression profiles of immune pathway genes, so as to investigate their performances in predicting survival of HCC patients and determine their clinical roles in existing staging models.

Materials and Methods

Data collection and processing

TCGA hepatocellular carcinoma RNA-Seq Counts data and clinical follow-up information concerning 306 tumor samples were downloaded from the HCCDB database (Lian et al., 2018) in November 5, 2018, finally 301 samples with stage information and followed up for more than one month were selected to be studied. Considering the age, gender, and stage distribution, we randomly divided the samples into two groups, namely, a training set (N = 158) and a test set (N = 143), as shown in Table 1. Furthermore, total 319 genes related to immune system process M13664, immune response M19817, immune effector process M14818 and immune system development and the M3457 pathway were downloaded from MsigDB database. We performed log2 conversion on the Counts data and standardized the data using EBSeq (Leng et al., 2013) to obtain an expression profile. The GSE14520 (Roessler et al., 2010; Roessler et al., 2012) standardized expression profile data were obtained from the GEO database, and a total of 221 samples with follow-up information were screened as an external validation set.

Analysis of immune characteristics between high-stage and low-stage

In order to identify the difference in expressions of immune genes at different Stages, Stage I and II were seen as the Low Stage group, while Stage III and IV were seen as the high stage group, the gene expression differences between the two groups were analyzed using the R software package EBSeq (Leng et al., 2013). Further genetic Fold Change ranked as a row. GSEA (Subramanian et al., 2007) was used for immunological pathway enrichment analysis, while t-SNE30 (Li et al., 2017) was used to analyze the distribution of genes in the four immunological pathways at high and low stages.

Univariate Cox proportional hazards regression analysis

As shown by Guo et al. (2018), univariate Cox proportional-hazards regression analysis was performed on each of immunized genes to screen genes significantly associated with OS in the training data set. P < 0.001 was considered as the threshold.

LASSO regression analysis

LASSO is a widely used regression model with many potential prognostic features, as it can perform automatic feature selection that create signatures with generally good prognostic performance (Kostareli et al., 2016). LASSO method has been extended to Cox model for survival analysis, and has been successfully applied for the purpose of building sparse signatures for survival prognosis in many application areas, including oncology (Papaemmanuil et al., 2013; Yuan et al., 2014; Zhang et al., 2013). In the current study, the R package glmnet (Friedman, Hastie & Tibshirani, 2010) was used for robust prognostic feature screening of immune genes, and optimal features were evaluated by ten-fold cross-validation.

Table 1 Random grouping information of TCGA samples.

		Traning set	Test set	
Age	≤60	80	73	
>60	78	70	
Gender	Male	110	96	
Female	48	47	
Stage	High	41	40	
Low	117	103	
Total	158	143	

Construction of prognostic immune gene signature

To obtain robust prognostic feature of immune gene, we performed a multivariate Cox regression analysis and constructed the following risk scoring model: RiskScore= ∑k=1nExpk∗ekHR

N refers to the number of prognostic immuno-genes, Expk is the expression value of prognostic immuno-genes, while ekHR is estimated regression coefficient of immuno-genes in the multivariate Cox regression analysis.

Correlation analysis between immune gene signature, GO and KEGG pathways

R software package GSVA (Hanzelmann, Castelo & Guinney, 2013) was used for analyzing GO and KEGG pathways of ssGSEA in each sample, and Pearson correlation between immune gene signature and sample pathway enrichment score was further analyzed. Finally, GO Term and KEGG pathways that were the most related to immune gene signature were determined.

RNA isolation and real-time PCR

The HCC cells culture medium in each well was aspirated as much as possible. 1 ml of Trizol (Invitrogen, Carlsbad, California) was added to the cells. The cells were placed horizontally for a while and blow evenly. The cells containing the lysate were transferred to a 1.5 ml EP tube and kept at room temperature for 5 min. 200 µl of chloroform was added to each tube and inverted for 15 s. After emulsification, let the tube stand for 5 min. After centrifugation at 12,000 g at 4 °C for 15 min, the upper aqueous phase was pipetted into a new 1.5 ml of EP and an equal volume of isopropanol (about 400 µl) was added to each tube and allowed to keep at room temperature for 10 min. After another centrifugation at 12,000 g at 4 °C for 15 min, the supernatant was discarded and 1 ml of pre-cooled 75% ice ethanol was added. After centrifugation at 7,500 g at 4 °C for 10 min, the supernatant was discarded. An appropriate amount of DEPC (20 µl) was added to dissolve the RNA. The purity and concentration of RNA was tested by 260 nm/280 nm using the NanoDrop nd-1000 spectrophotometer (NanoDrop Technologies, Wilmington, DE, USA). According to the program provided by the manufacturer (Thermo Fisher Scientific, Waltham, USA), reverse transcription cDNA kit was used to reverse transcribe 1 microgram total RNA for Synthesis of cDNA (42 °C for 60 min, 70 °C for 5 min, 4 °C preservation). SYBR Green PCR Master Mix (Roche, Basle, Switzerland) was used to perform quantitative real-time polymerase chain reaction (RT-qPCR) experiment using Opticon real-time PCR Detection System (ABI 7500, Life technology, USA). The PCR cycle was as follows: pretreatment at 95 °C for 10 min; followed by 40 cycles of 94 °C for 15 s, 60 °C for 1 min, finally at 60 °C for 1 min and at 4 °C for preservation. The relative mRNA quantity was determined using the comparative cycle threshold (ΔΔCt) method. GAPDH expression was used for normalization. The primer sequences were used for RT-qPCR analysis as Table S1.

Statistical analysis

Kaplan–Meier (KM) curve was plotted, with the median risk score in each data set used as a cutoff in comparing the risk of survival between the high risk group (HC) and the low risk group (LC). Multivariate Cox regression analysis was performed to determine whether the immune-genic marker is an independent prognostic factor. Statistical significance was defined as P < 0.05. All analyses were performed in R 3.4.3, and all analyses were based on default parameters unless otherwise stated.

Results

Different immunological characteristics exist in high stage and low stage

Among the 301 samples obtained from TCGA, 220 cases of Stage I and II were found in the LC group, while 81 cases of Stage III and IV were in the HC group. The GESA analysis results showed that the LC group was in the immune response M19817, immune system process M13664, and immune effector process M14818 pathway activity in the LC group was higher than that in HC group (Figs. 1A–1C). The expressions of 319 immune-related genes in all liver cancer samples was subjected to dimensionality reduction analysis using T-SNE, and the results demonstrated that the HC group was more concentrated compared with the LC group, indicating that these immune genes have different distribution characteristics (Fig. 1B). Thus, HC and LC had different immunological characteristics.

Figure 1 Different immunological characteristics exist in high stage and low stage.

(A–C) GSEA enrichment for immune response, immune system process and immune effector process pathway, na-pos means HC group, na-neg means LC group, three pathways for FDR < 0.001. (D) The expression of 319 immune-related genes were conducted to Dimensional analysis of T-SNE. Red indicates the LC group and black indicates the HC group.

Immune gene can be used as a prognostic marker

Cox proportional hazard regression model (Kneale, Mancuso & Stewart, 1981) was used for performing a univariate analysis of the relationship among 319 immune genes and HCC prognosis, as shown in Table 2, 17 genes were identified as significantly associated with prognosis, specifically, 4 genes were risk factors, while 13 genes were protective factors. Moreover, most of these genes were found related to complex diseases, for example, the most significant homozygous deletion of CFHR1 is associated with leukemia (Fratelli et al., 2016), APOA2 allelic imbalance is involved in endometrial cancer (Goumenou et al., 2001), and specific inhibition of SOD1 selectively promotes cancer cell apoptosis through regulating the ROS signaling network (Li et al., 2019). Furthermore, we performed a multivariate regression analysis to establish a risk model for 17 immune genes with the model’s AUC = 0.87 (Fig. 2A). According to the proportion of the cases in HC and LC, 70% quantile was taken as the threshold value of high-risk and low-risk grouping, that is, cutoff = −12.808. Significant prognostic differences between high and low group samples were identified (Fig. 2B). Similarly, in the test set AUC = 0.88 (Fig. 2C), the prognosis of the high-risk group was significantly worse than that of the low-risk group (Fig. 2D). In conclusion, immune genes have great potential as prognostic markers.

Table 2 Seventeen prognostic immune genes.

Genes	SYMBOL	lnHR	HR (95%_CI_for_HR)	wald.test	p.value	
3078	CFHR1	−0.26	0.77 (0.69–0.85)	25	6.90E−07	
336	APOA2	−0.24	0.79 (0.72–0.87)	24	9.60E−07	
717	C2	−0.49	0.61 (0.5–0.75)	24	1.20E−06	
6647	SOD1	−0.93	0.4 (0.27–0.58)	22	3.30E−06	
7448	VTN	−0.28	0.75 (0.67–0.85)	21	4.40E−06	
7494	XBP1	−0.94	0.39 (0.25–0.6)	18	2.30E−05	
6288	SAA1	−0.19	0.83 (0.76–0.91)	17	3.70E−05	
1394	CRHR1	0.62	1.9 (1.4–2.5)	15	8.80E−05	
335	APOA1	−0.18	0.83 (0.76–0.91)	15	0.00011	
6480	ST6GAL1	−0.45	0.64 (0.5–0.8)	14	0.00014	
246778	IL27	−0.32	0.72 (0.61–0.86)	13	0.00031	
1510	CTSE	0.18	1.2 (1.1–1.3)	12	0.00044	
2217	FCGRT	−0.77	0.46 (0.3–0.72)	12	0.00064	
23054	NCOA6	1	2.8 (1.5–5.1)	11	0.00073	
85477	SCIN	0.24	1.3 (1.1–1.5)	11	0.00079	
7704	ZBTB16	−0.27	0.76 (0.65–0.89)	11	0.00081	
912	CD1D	−0.3	0.74 (0.62–0.89)	11	0.001	

Figure 2 Immune gene can be used as a prognostic marker ps://.

(A) AUC curve of 17 immune genes in the TCGA training set. (B) Kaplan-Meier curve of prognosis difference between high and low risk group of TCGA training group. (C) AUC curve of 17 immune genes in TCGA test set. (D) Kaplan-Meier curve of prognosis difference between high and low risk group of TCGA test group.

Identification of a 7-immu-gene signature for HCC survival

Seventeen prognostically significant immune gene expression profiles were used in LASSO for dimensionality analysis in the training set, and 10-fold cross-validation was selected. When λ = 0.02359167, the error rate was minimized (Fig. 3A), and seven genes, namely, IL27, CD1D NCOA6, CTSE, FCGRT, CFHR1 and APOA2, were obtained, among which IL27 plays an important role in cancer and immunity (Fabbi, Carbotti & Ferrini, 2017), CD1D is a key factor in immunotherapy (King et al., 2018) and the homozygous deletion of CFHR1 is associated with acute myeloid leukemia (Fratelli et al., 2016). A 7-immuno-gene signature was established by a multivariate COX regression analysis with the model: Risk7=−0.05983∗CFHR1−0.01437∗APOA2−0.10219∗IL27+0.10394∗CTSE−0.18521∗FCGRT+0.37195∗NCOA6−0.11573 ∗ CD1D.

Figure 3 Identification of a 7-immu-gene signature for HCC survival.

Lasso regression analysis results: (A) the trajectory of each independent variable, the horizontal axis represents the log value of the independent lambda, the vertical axis represents the coefficient of the independent variable. (B) the average error rate interval under each lambda. (C) The relationship between 7 immune genes and risk scores.

The risk score of each sample were calculated, the samples were grouped according to 70% quantile of the risk score, and the relationship between the expressions of 7 genes and the risk scores were analyzed. We found that as protective factors, the low expressions of CFHR1, APOA2, IL27, FCGRT and CD1D were correlated with high risk of developing poor HCC prognosis, whereas high expressions of risk factors CTSE and NCOA6 were related to high risk of poor prognosis (Fig. 3C). The prognosis was significantly different between the high-risk and low-risk groups (Fig. 4B), as in the training set, the 7-gene signature had an AUC of 0.87 (Fig. 4A), while that the AUC in the TCGA test set was 0.89 (Fig. 4C). Moreover, the prognosis of high-risk samples was significantly worse than that of low-risk samples (p = 0.0087, Fig. 5D). Similarly, by applying the model to all TCGA samples, AUC = 0.9 (Fig. 4E), and the prognosis of high-risk samples was significantly worse than that of low-risk samples p < 0.0001 (Fig. 4F). Consistent with the training set, the model was less effective in classifying prognosis result within the TCGA data set.

Figure 4 Identification of a 7-immu-gene signature for HCC survival 5.

(A) ROC curve of the TCGA training set. (B) Kaplan-Meier curve of high and low risk group prognosis difference in the TCGA training set. (C) ROC curve of the TCGA test set. (D) Kaplan-Meier curve of high and low risk group prognosis difference in the TCGA test set. (E) ROC curve of the dataset of all samples of TCGA. (F) Kaplan-Meier curve of prognosis difference between high and low risk groups in the dataset of all samples of TCGA.

Figure 5 Model performance evaluation ashes.

(A) ROC curve of the external verification data set. (B) Kaplan-Meier curve for prognosis differences between high and low risk groups in the external validation data set. (C) ROC curve of the model in the TCGA data set for 1 year, 3 years, and 5 years. (D) AUC of the Guo et al. (2018) model in the TCGA data for 1 year, 3 years and 5 years.

Model performance evaluation

The external dataset GSE14520 data was used for validating our model. The model was applied to the GSE14520 data, and the risk score for each sample was calculated as AUC = 0.66 (Fig. 5A). The samples are grouped into high and low risk groups according to their medians. The prognosis in the high-risk group was found significantly worse than that in the low-risk group, with p < 0.001 (Fig. 5B), indicating that the model was applicable for risk stratification of data from different platforms. Furthermore, by analyzing a recently discovered 8-gene signature (Qiao et al., 2019), we compared the AUC in our model with that in Guo et al.’s (2018) in the TCGA data for 1-, 3- and 5-year prognosis (Figs. 5C, 5D), and the results demonstrated that AUC in our model showed a better performance than the previous model.

Immune gene signature, GO and KEGG pathway analysis

Risk score of each sample was calculated based on the immune gene signature in the test sample, and the ssGSEA was used for analyzing GO and KEGG pathways. The Pearson correlation between the immune gene signature and the sample pathway enrichment score was also calculated. 5 GO term and KEGG Pathway with the highest correlation were selected for comparing them with the sample analysis scores (Fig. 6), and we observed a negative correlation between the PPAR signaling pathway and the sample risk value in the signaling pathway, while prion diseses, huntingtons disease, preroxisome and amyotrophic leteral sclerosis als pathway were positively correlated with sample risk. Moreover, in GO Term, all terms had a negative correlation with sample risk, including multiple immune system processes. To conclude, immune gene signatures are associated with pathways that are involved in a variety of complex diseases.

Figure 6 Immune gene signature and GO and KEGG pathway analysis, horizontal representation of samples, heat map representation of GO, KEGG pathway enrichment scores.

The expression levels of genes in MHCC97

We have demonstrated that 7 immune-related genes can be used as prognostic indicators for hepatocellular carcinoma, and further detected the expression of 7 genes on hepatocellular carcinoma MHCC97 cells by RT-qPCR assay. We found that the levels of CFHR1, APOA2, IL27, FCGRT and CD1D were significantly downregulated in MHCC97 cells, while the expressions of CTSE and NCOA6 were obviously upregulated in MHCC97 cells (Fig. 7). Those results indicated that those genes were associated with development of hepatocellular carcinoma.

Figure 7 The expressions of genes in MHCC97.

The levels of CFHR1, APOA2, IL27, FCGRT and CD1D were significantly downregulated in MHCC97 cells, while the expressions of CTSE and NCOA6 were obviously upregulated in MHCC97 cells.

Discussion

In recent years, tumor immunotherapy has achieved remarkable success in the treatment of advanced tumors (Maccio & Madeddu, 2012; Sharma et al., 2011). A comprehensive understanding of liver cancer requires not only focus on tumor cells, but also on tumor microenvironment (Leonardi et al., 2012; Yarchoan et al., 2017), that contains a variety of cell populations interacting with cancer cells and different stages of tumorigenesis. The tumor-infiltrating immune cells and immune response in tumor microenvironment have attracted much research attention, and has become a promising therapeutic target. Further research into in what ways immune signatures are associated with HCC tumorigenesis and progression can contributes to the development of novel and specific targeted therapeutic strategies, especially in the setting of combined therapies. In the current study, the differences in immunological characteristics between the HC group and the LC group were analyzed, and we found that the immune response M19817, immune system process M13664, immune effector process and M14818 pathway were more active in LC group than that in the HC group in, suggesting that different tumor stages had different immune characteristics.

As HCC patients at similar TNM stages have different survival times, HCC is seen as a highly heterogeneous disease for its prognosis. As liver cancer is increasingly detected and treated in its early stages, traditional clinicopathological indicators such as tumor size, vascular invasion, portal vein tumor thrombus, and TNM staging could no longer satisfy the current needs in predicting individual outcomes. With the development of gene expression characteristics such as Oncotype DX (Siow et al., 2018) and Coloprint (Tan & Tan, 2011), it has been shown that gene expression profiling in screening new prognostic markers in cancers has become the most promising method for high-throughput molecular identification. Guo et al. (Qiao et al., 2019) have developed an 8-gene signature with an AUC around 0.7; however, the current study used an immune gene set to construct 7- gene signature, with AUC reached 0.89, which suggested that our immune gene had higher predictive value as a prognostic marker.

Our 7-immuno-gene signatures CFHR1, APOA2, IL27, FCGRT, and CD1D were protective factors, while CTSE and NCOA6 were risk factors. Urinary quantification of APOA2 protein was decscribed, which is a biomarker for the diagnosis of bladder cancer (Chen et al., 2015), and APOA2 had higher expression in liver tissues (Fukuhara et al., 2014), which differ from our study, consider cell heterogeneity. Shi et al. (2016) found that the level of FCGRT was down-regulation in clinical HCC samples. It has been reported that IL27 plays an important role in cancer and immunity (Fabbi, Carbotti & Ferrini, 2017), CD1D is a key factor in immunotherapy (King et al., 2018), and the homozygous deletion of CFHR1 is associated with acute myeloid leukemia (Fratelli et al., 2016), moreover, the stratified analysis of expression and prognosis found that NCOA6 is highly positively correlated with the grade of liver cancer, and this was the first time to prove that NCOA6 is associated with clinical stratification of liver cancer. Knockdown of signature gene CTSE for CD24+/CD44+ cells significantly reduced self-renewal ability on HCC cells in vitro and in vivo (Ho et al., 2019). Meanwhile, our results of GSEA analysis indicated that the 7-immuno-gene signature-enriched pathway was significantly associated with tumorigenesis pathways and immune-related biological processes. These results indicate that the model developed in the current study has potential clinical application value, and can provide a potential target for the diagnosis of HCC patients.

Though bioinformatics techniques was applied to identify potential candidate immunogene markers involved in tumorigenesis in large samples, the current study still had certain limitations, for example, our samples lacked clinical follow-up information, thus, factors related to other health status of the HCC patients were not taken into investigation as prognostic biomarkers. Moreover, our results were obtained only through bioinformatics analysis, therefore, requires further experimental validations.

Conclusion

In conclusion, the current study systematically explored the immunogenic gene expression profiles in different aspects of liver cancer. We found 7-immuno-gene features in HCC, and had higher AUC value in both training and validation sets. Compared with clinical features, gene classifiers can improve survival risk prediction. Therefore, the classifier identified in the study could be used as a molecular diagnostic test in assessing the prognostic risk of HCC patients.

Supplemental Information

Table S1 Primers for RT-qPCR

Click here for additional data file.

Additional Information and Declarations

Competing Interests

Author Contributions

Data Availability

The authors declare there are no competing interests.

Feng Xue and Lixue Yang performed the experiments, authored or reviewed drafts of the paper, and approved the final draft.

Binghua Dai performed the experiments, analyzed the data, and approved the final draft.

Hui Xue performed the experiments, analyzed the data, prepared figures and/or tables, and approved the final draft.

Lei Zhang performed the experiments, prepared figures and/or tables, and approved the final draft.

Ruiliang Ge conceived and designed the experiments, prepared figures and/or tables, authored or reviewed drafts of the paper, and approved the final draft.

Yanfu Sun conceived and designed the experiments, authored or reviewed drafts of the paper, and approved the final draft.

The following information was supplied regarding data availability:

Data is available at TCGA (search term: TCGA-LIHC, https://portal.gdc.cancer.gov/projects/TCGA-LIHC), NCBI GEO (GSE14520), and HCCDB (HCCDB15; http://lifeome.net/database/hccdb).

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
