# Peer review of "Bioinformatics profiling identifies seven immune-related risk signatures for hepatocellular carcinoma"

_PeerJ, doi:10.7717/peerj.8301_

## Round 0.1 · original submission · Major Revisions

Please address the concerns raised the reviewers.

·

Basic reporting

English should be carefully checked by native English speakers. Figures and Tables (Results) support their hypothesis well. Literature summary regarding the impact of expression of CFHR1, APOA2, IL27, FCGRT, CD1D, CTSE, and NCOA6 should be done and discussed.

Experimental design

In this paper, research question is well defined, and it is significant in the research field. The knowledge gap being investigated is well explained and the authors clarified how they contribute to filling that gap. Methods are well described in this study with adequate information and references.
However, I strongly recommend them to perform functional analysis using cell lines (and siRNAs) to validate accuracy the markers (CFHR1, APOA2, IL27, FCGRT, CD1D, CTSE, and NCOA6) they identified as immune-related signatures for HCC in this study. If this proposed experiment is hard, the authors should compare the expression level of these genes in different differentiation stages of liver cancer cell lines (differentiated vs undifferentiated).

Validity of the findings

This study provides reproducible analysis and its impact is important for the field of liver cancer research. The conclusion is well illustrated, and it is closely related to their hypothesis.

Additional comments

The author bioinformatically analyzed the transcript sequencing data of liver cancer to find an immune gene marker to predict the clinical outcome of HCC using TCGA database. As such, they concluded 7 immunogenic genes as novel prognostic markers for predicting survival in patients with HCC. This paper is well written, and their analysis is clear. However, functional studies (as suggested) should be performed to validate the bioinformatic analysis.

Reviewer 2 ·

Basic reporting

The manuscript is clear and well structured. English written is OK, and if get more improvement will better

Experimental design

The results of this paper are potentially interesting and it appears that the data are in agreement with the conclusions of the authors. The bioinformatic analysis method and statistical analysis used appropriately and described with sufficient detail.

Validity of the findings

Conclusions are well stated, linked to original research question .

Additional comments

The results of this paper are potentially interesting and it appears that the data are in agreement with the conclusions of the authors. The bioinformatic analysis method and statis used appropriately and described with sufficient detail. However, the discussion of manuscript does need strengthen, what are the relevant and meaningful among 7 immune genes? Any possible suggestions in clinic from authors? if authors concluded them as biomarkers for diagnosis? Though authors could not provide the experimental validation for 7 immune signature found in the current study, more related report in liver cancer or other type of cancer of those 7 genes should be discussed in Discussion.

---

## Round 0.2 · accepted · Accept

Thanks for allowing us to consider the manuscript